# A Shot in the Arm for Vaccination Intention: The Media and the Health Belief Model in Three Chinese Societies

**DOI:** 10.3390/ijerph19063705

**Published:** 2022-03-20

**Authors:** Ruoheng Liu, Yi-Hui Christine Huang, Jie Sun, Jennifer Lau, Qinxian Cai

**Affiliations:** Department of Media and Communication, City University of Hong Kong, Hong Kong 999077, China; rhliu9-c@my.cityu.edu.hk (R.L.); jsun27-c@my.cityu.edu.hk (J.S.); jennifer.lau@my.cityu.edu.hk (J.L.); qinxicai-c@my.cityu.edu.hk (Q.C.)

**Keywords:** vaccination intention, media exposure, health belief model, media trust, Chinese societies

## Abstract

This large-sample study of three Chinese societies—Mainland China, Taiwan, and Hong Kong—demonstrates the importance of media exposure for people’s vaccination intentions during the COVID-19 pandemic. By employing two constructs (i.e., perceived susceptibility and severity) in the health belief model (HBM), the study identifies significant indirect effects of media exposure on individuals’ vaccination intention in all three Chinese societies. That said, media trust negatively moderated the path from perceived severity to vaccination intention in Mainland China and Taiwan. In these two societies, the higher an individual’s trust in media, the less influence of perceived severity on his/her vaccination intention. It suggests that the level of trust in media is a contextual factor in explaining individuals’ decision-making on health issues. Generally, the combination of the HBM and media trust has been proven to be useful for understanding individuals’ vaccination intentions. These findings provide practical considerations for governmental agencies, public institutions, and health campaign designers to promote vaccination in the pandemic.

## 1. Introduction

Vaccines, proven to be safe, effective, and life-saving, provide a high degree of prevention against severe illnesses and deaths from COVID-19 [1]. The premise for ending this pandemic is to have a large share of the world immune to the virus, for which vaccination is the safest way [2]. Human beings have long relied on vaccines to bring down the death rate of infectious diseases (e.g., influenza, hepatitis [3,4]), given that the effectiveness of vaccination has been frequently and widely studied and verified [5,6]. Recently, one study [7] demonstrated that the effectiveness of a specific type of vaccine—Pfizer—against infections of the COVID-19 delta variant was 93% during the first month after full vaccination and remained at 53% after four months. Similarly, the effectiveness of vaccines was also proven to resist the delta variant—the effectiveness of two doses with the BNT162b2 vaccine was 88%, and 67% with the ChAdOx1 nCoV-19 vaccine [8].

Unfortunately, in spite of the scientific evidence of the validity of COVID-19 vaccination [9] and persistent calls from the World Health Organization [10] to the public, the response was not satisfactory [11]. Nevertheless, as of 7 October 2021, less than half the world population had received at least one dose of the COVID-19 vaccines (i.e., 46.1%, [12]). The World Health Organization [13,14] has already identified vaccine hesitancy—the reluctance to receive vaccines despite the availability of vaccines—as a significant barrier to overcome. Thus, to improve the vaccination rates and diminish the death toll of the COVID-19 pandemic, it is essential to understand the factors and underlying mechanisms contributing to an individuals’ intention to get vaccinated.

Numerous factors may influence people’s vaccination intention, such as the origin of the vaccines, the qualification of the vaccine manufacturers, people’s education level [15]. People approach vaccine-related information from different sources, one of which is the media. Accordingly, this study investigated the importance of media during the pandemic using the variables of media exposure and media trust. Media are empirically identified as the amplifier of the social amplification process [16,17], in which the perception of technical risks or threats and their negative effects on individuals’ well-being are increased [18]. Specifically, social scientists have emphasized the significant role media play in individuals’ vaccine behaviors (e.g., influenza vaccine [19], HIV vaccine [20], HPV vaccine [21], childhood vaccination [22]). One of the related constructs, media exposure, tends to have an indispensable effect on the vaccination intention among individuals (see [23] for a systematic review). Specifically, a study regarding COVID-19 vaccines revealed that vaccine-related information on social media would negatively predict individuals’ attitudes toward vaccination [24]. We also examined another construct, media trust, which generally acknowledges that media, as a typical institution on which people could rely [25], may motivate people to initiate health behaviors [26]. This study aimed to further examine whether the level of trust in media was context-related during the pandemic (e.g., related to the relative severity of the pandemic).

With media exposure and media trust in mind, we developed a theoretical framework that leveraged the health belief model (i.e., HBM, [27]) to examine the psychological mechanism behind individuals’ decision-making process for vaccination uptakes. The perceived susceptibility and perceived severity factors of HBM have been demonstrated to serve a mediating role between media exposure and vaccination intention. However, some scholars [28,29] have questioned the utility of these two constructs in that both have presented little predicting power when examined in certain contexts. Thus, this study aims to examine the validity of such assertions. Additionally, the COVID-19 pandemic has led to increased global interest in the Chinese experience [30]. Researchers have indicated that Confucian values have contributed to the achievement Chinese societies have made in their COVID-19 response [31,32]. Thus, this study will explore the general pattern of media effect on individuals’ vaccination intention in three typical Chinese societies—Mainland China, Taiwan, and Hong Kong [33]. Although different in certain aspects (e.g., political systems) [34], these three societies are all deeply embedded in Confucian culture. While acknowledging and understanding the dimensions of media power, it will provide practical guidance for government institutions and vaccination programs on how to utilize media efficiently and effectively. Specifically, this study hypothesized a framework for the effect of media exposure on vaccination intention based on HBM (Figure 1).

### 1.1. Media Exposure

According to Slater [35], media exposure is defined as the amount to which audiences have been exposed to specific messages or message types. Social scientists have emphasized that media plays a crucial role in shaping individuals’ health-related behaviors [36]. Specific exposure to media has encouraged individuals’ motivation to pursue desirable health outcomes [37,38] and to prevent them from pursuing undesirable ones [39]. For instance, Pearl and colleagues [40] found a significant correlation between exposure to weight-stigmatizing video and exercise intentions/behaviors among females. A certain amount of media exposure may also motivate individuals to take precautions to prevent various diseases (e.g., cancer prevention, Prochaska, [41], HIV prevention [42]), and to stay away from tobacco use, drug-taking, and alcohol use (see [43] for a meta-analysis). Researchers have also demonstrated social media was prominent in preventing the public from suffering infectious diseases [44].

The significant influence of media exposure to vaccination intention has already been confirmed by social scientists (see [45] for a systematic review). In an empirical study concerning HPV vaccines, media coverage of political and gendered messages educated individuals’ understanding of vaccines [46]. With an online survey, Lin and Lagoe [47] found that college students’ news media dependency would positively influence their vaccination intention during the H1N1 pandemic.

Upon the current pandemic, a plethora of worldwide studies have also started to investigate the link between media exposure and COVID-19 vaccination behaviors [48,49]. In China, it was found that netizens’ social media use would predict their intention to take preventive measures, including vaccination [50]. In Germany, Gehrau and his colleagues [51] also revealed that besides information from authorities/experts, local newspapers also positively influenced individuals’ COVID-19 vaccination intention. In Taiwan, other than health professionals and governmental agencies, mainstream and social media were also proven to be effective sources to providers of pandemic and vaccine-related information to the public, which could consequently promote people’s vaccination intention [52]. With these empirical supports, we hypothesized that individuals exposed to more media coverage about the pandemic would be more likely to receive vaccines, as follows (Figure 1):

**Hypothesis** **1** **(H1).**
*Media exposure to COVID-19 will positively influence individuals’ vaccination intention.*


### 1.2. Perceived Susceptibility and Perceived Severity in the HBM

Perceived susceptibility and severity are frequently combined and labeled as perceived threats [53] since both constructs concern threat. Perceived susceptibility generally refers to individuals’ beliefs on the probability of contracting an illness; perceived severity focuses on their perceptions of how serious such an outcome could be [54].

These two constructs are also two important elements in the classic health belief model (i.e., HBM, [27]). Aiming to understand individuals’ failure to take preventive health actions [27], the HBM proposed that individuals’ actual health behaviors can be predicted by (1) perceived susceptibility and (2) perceived severity of certain diseases, as well as the (3) perceived benefits and (4) perceived barriers of the behavior [27]. Perceived benefits and barriers refer to individuals’ beliefs on the advantages and disadvantages of certain behavior [55,56].

To elaborate the difference between the two constructs related to threat, perceived susceptibility paid close attention to the possibility for individuals to experience a disease, while perceived severity emphasized the general consequences of the disease [57]. While scholarship produced different standards and inconsistent results, perceived susceptibility and severity beliefs were generally acknowledged to present the slightest predicting power on outcomes in HBM [56]. According to Janz and Becker’s meta-analysis [58], perceived barriers were the most potent predictor of behaviors, while perceived severity was the least potent. In contrast, Harrison and colleagues [28] revealed a comparatively small impact of each variable in their meta-analysis study. More recently, Carpenter [29] conducted a more rounded meta-analysis that confirmed that perceived severity remained the least powerful construct and that the link between perceived susceptibility and outcomes was close to zero. 

Nevertheless, other studies have supported the viability of both constructs in facing disease contexts [59,60]. A great majority of people first assessed their susceptibility and severity, other than perceived benefits and barriers, when encountering certain diseases [27]. 

The relationship between individuals’ media exposure to health-related information and their perceptions of susceptibility and severity has been widely reviewed and confirmed in a health context [61,62]. Zhang and Zhou’s [63] study concluded that health risk messages would shape individuals’ perception of threat (i.e., perceived susceptibility and perceived severity). Specifically, researchers also revealed a significant correlation between social media use and individuals’ perceived threat of the coronavirus in the COVID-19 pandemic [64]. Similarly, previous studies also indicated a close relationship between perceived threats and individuals’ vaccination behaviors [65,66]. For instance, a recent study demonstrated a correlation between the perceived threat of the COVID-19 pandemic and individuals’ influenza vaccination [67]. Likewise, a cross-sectional survey of health care students in China showed that students who perceived COVID-19 as a serious disease were more likely to get vaccinated [68]. 

Scholars have identified many theoretical constructs as mediators in the relationship between media exposure and health behaviors (e.g., general health behaviors, [69]; emotional responses, [70]). The HBM has also been frequently adopted to study the indirect relationship between media use and behavioral health outcomes (see [71] for a systematic review) and has been shown the utility of explanation and prediction [58].

Just as Fishbein and Cappella [72] asserted the necessity of reviewing the mediator mechanism through which health-related information is processed, we hypothesized that individuals exposed to more media coverage of the pandemic would tend to perceive more susceptibility to and severity of the disease, which would, in turn, heighten their vaccination intention (Figure 1): 

**Hypothesis** **2** **(H2).**
*The relationship between media exposure to COVID-19 and individuals’ vaccination intention will be mediated by (a) perceived susceptibility and (b) perceived severity of the disease.*


### 1.3. Media Trust 

Trust has frequently been considered an activator for relationships in various disciplines (e.g., business, communication, information management, and sociology [73]). Trust is essential for understanding interpersonal information processing behaviors [74] as it establishes in society a great level of certainty, confidence, and predictability [75]. Various empirical studies have viewed the moderating role of trust on several health-related behaviors [76,77,78]. For example, Lee and Hornik [79] indicated that the relationship between Internet use and physician visits in the United States was positively moderated by patients’ trust in physicians. 

Trust in media was defined as the degree to which people perceived media to be capable of doing their job and their concern with the public and social interests [25,80]. Three categories of media trust were documented in the past literature: (1) trust in media institutions, (2) trust in media disseminators, and (3) trust in media content [81]. People trust media institutions and disseminators first and then trust their content [82]. Therefore, the current study will examine media trust from the perspective of trustees (i.e., media institutions, such as government media, and media disseminators, such as opinion leaders on media) instead of the media content.

We assumed that people who did not trust a type of media would be less influenced by that media and would tend to seek other information sources [83]. Within communication scholarship, media trust has been a typical construct for examining media consumption but has received increasing research interest from scholars in other social science disciplines to explore its effect on people’s cognitive behaviors [83,84].

Scholars previously have elucidated that people are motivated to commit related health behaviors promoted by media that they trust [26,85]. Tokuda, Fujii, Jimba and Inoguchi [84] illustrated that Asians’ trust in mass media (i.e., TV and newspaper) was associated with their health because they tended to believe media’s health-related information and to follow media recommendations. More recently, Wu and Shen [86] discovered that in the COVID-19 pandemic, people’s compliance with health practices was reinforced by media trust. Research also has indicated that media trust predicted individuals’ attitudes toward vaccination; particularly with a lower level of trust in media, people would tend to reject the idea of getting vaccinated [87]. 

Empirical studies also proved the effect of media use or exposure on individuals’ trust with institutions and their personnel (e.g., political and social trust [88], trust in scientists [89], trust in government [90]). People’s media exposure was also correlated with their trust in media [91,92].

Carpenter’s meta-analysis [29] contended that the mediator/moderator should be involved in the relationship between perceived susceptibility/severity and actual behavior and called for further exploration. Some scholars answered that invitation. For instance, Witte [54] put forth the extended parallel process model (i.e., EPPM), in which the effects of perceived susceptibility and severity on behaviors were moderated by self-efficacy. 

To support Carpenter’s proposition on mediator/moderator role involvement, this study explored media trust as the moderator. Some scholars have already indicated the interaction effect of individuals’ perceptions of threat, which is highly related to susceptibility and severity as previously explained, and perceived trust in media on their initiation of health behaviors (e.g., social distancing behavior amid COVID-19 [93]). Lin and Bautista [94] also confirmed such a relationship regarding individuals’ protective measures during haze. Therefore, we hypothesized that individuals’ trust in media would influence their vaccination intention by moderating the impacts of perceived susceptibility and perceived severity (Figure 1):

**Hypothesis** **3** **(H3).**
*Media trust will moderate the mediating influence of (a) perceived susceptibility and (b) perceived severity on the effect of COVID-19 vaccination intention.*


## 2. Subjects and Methods

### 2.1. Participants and Procedure

Online surveys were conducted in Mainland China, Taiwan, and Hong Kong from August 2020 to January 2021. Independent agencies Rakuten Insights in Mainland China and Hong Kong and the Chungliu Education Foundation in Taiwan administered the process of participant recruitment. 

In Mainland China (August to September 2020), a combination of Probabilities Proportional to Size (PPS) sampling and quota sampling with gender and age was adopted based on the 2010 Mainland population census. In Hong Kong (September to October 2020), stratified quota sampling was utilized based on the distribution of gender, age, and residence in the 2016 Hong Kong population census. In Taiwan (October 2020 to January 2021), we used Random Digit Dialing (RDD) sampling with the Taiwan 2020 household registration database. All participants in the three regions were invited to fill in a web-based questionnaire on Qualtrics (in Mainland China and Hong Kong) and SurveyCake (in Taiwan). The survey was completed with a total of 9637 valid responses (Mainland China: *n* = 3389; Taiwan: *n* = 3000; Hong Kong: *n* = 3248).

Upon the data collection period, the COVID-19 situation in the three societies was not severe. The daily number of confirmed cases was less than 1 case per million people in Mainland China, Taiwan, and Hong Kong [12]. Furthermore, the Delta variant that started in late 2020 in India had not spread into the three societies at that moment [95].

To ensure validity and reliability of the questionnaire, pretests were conducted with ten students in each area. Back-translation [96] was performed to make sure the original English questions were correctly conveyed. Moreover, using the software G*Power 3 [97], a post-doc power analysis was conducted. The sample sizes of 3389 (Mainland), 3000 (Taiwan), and 3248 (Hong Kong) were used for the power analyses, and four predictor variables were used as baselines. The results showed that the power for these three samples all reached 1.0, signifying the greatest power.

### 2.2. Demographics of the Sample

Table 1 shows detailed demographic profiles (i.e., gender, age, and education) of the samples in the three societies. Mainland China was close to 1:1 gender proportion while Taiwan and Hong Kong had more female respondents (Taiwan 1:1.278 and Hong Kong 1:1.128). Around 53–59% of subjects in Hong Kong and the Mainland were aged over 40, while Taiwan had a younger sample with only 29.7% in that older age group. The means of monthly household income were calculated. 

Participants in Hong Kong had the highest income with USD 3857–5142, while those in Taiwan had the lowest with USD 1424–1780. Participants in Mainland China had the average income of USD 2330–3885. Among all three societies, the sample in Taiwan was the most highly educated, with 71% of the respondents having at least a bachelor’s degree, compared to Hong Kong’s percentage of only 35.3%.

Regarding the residential areas, participants were recruited from 26 provinces or municipalities in Mainland China. The top three provinces of participants were Guangdong Province (7%), Anhui Province (6.8%), and Jiangsu Province (6.8%). For Taiwan, participants were recruited from 22 cities or counties, among which New Taipei City (20%) and Taipei City (14.5%) were the top two cities. In Hong Kong, half of the participants were from New Territories (50%), and 32.1% from Kowloon, followed by 17.5% from Hong Kong Island. 

### 2.3. Measures

Media exposure to the COVID-19 pandemic was measured by the tendency of individuals to be exposed to COVID-19 pandemic information on respective media, including television, newspapers, media-sharing websites (e.g., Sina, Beijing, China), social network sites (e.g., WeChat, Tencent, Shenzhen, China), and short video platforms (e.g., Douyin, ByteDance, Beijing, China) on the Mainland; television, newspapers, media-sharing websites (e.g., Yahoo, Sunnyvale, CA, USA), social network sites (e.g., Facebook, Menlo Park, CA, USA), and short video platforms (e.g., Tik Tok, ByteDance, Beijing, China) in Taiwan; and television, newspapers, media-sharing websites (e.g., Yahoo), social network sites (e.g., Facebook), and short video platforms (e.g., Tik Tok) in Hong Kong. Answers to the questions were recorded on a seven-point Likert scale (1 = very little to 7 = very much). The responses were then averaged to create media exposure to COVID-19 pandemic (Mainland: M = 4.45, SD = 1.22, Cronbach’s α = 0.85; Taiwan (TW): M = 3.70, SD = 1.01, Cronbach’s α = 0.69; Hong Kong (HK): M = 3.73, SD = 1.07, Cronbach’s α = 0.74).

The perceived susceptibility to COVID-19 was measured with three questions rated on a seven-point Likert scale ranging from 1 = strongly disagree to 7 = strongly agree [55,98]. The question items included “The coronavirus is almost ubiquitous, and the pathogenicity is high”, “There is a high probability of getting infected”, and “The mortality rate of COVID-19 is high”. The responses were then averaged to create perceived susceptibility to COVID-19 (Mainland: M = 4.78, SD = 1.31, Cronbach’s α = 0.81; TW: M = 4.54, SD = 1.08, Cronbach’s α = 0.78; HK: M = 4.45, SD = 1.12, Cronbach’s α = 0.79).

The perceived severity of COVID-19 was measured with four question items on a seven-point Likert scale (1 = strongly disagree to 7 = strongly agree) adopted from Champion [98]. The question items included “COVID-19 has had a major influence on the economy/politics/people’s lives/society”. The responses were then averaged to create the perceived severity of COVID-19 (Mainland: M = 5.18, SD = 1.20, Cronbach’s α = 0.88; TW: M = 5.22, SD = 1.19, Cronbach’s α = 0.91; HK: M = 5.50, SD = 1.06, Cronbach’s α = 0.88).

Media trust was measured on a seven-point Likert scale (1= strongly distrust to 7 = strongly trust) based on previous empirical studies [83,84]. The question items in Mainland included trust in central government media, local government media, commercial media, opinion leaders on WeChat or Weibo, and opinion leaders on TV/newspaper (M = 5.28, SD = 0.90, Cronbach’s α = 0.79). Items in Taiwan included trust in government media, opinion leaders on the internet, and opinion leaders on TV/newspaper (M = 4.25 SD = 1.09, Cronbach’s α = 0.82). For Hong Kong, trust in government media, opinion leaders on the Internet, and opinion leaders on TV/newspaper were listed in questionnaires (M = 3.76, SD = 1.11, Cronbach’s α = 0.76).

Vaccination intention was adapted from Yang and Pittman [99]. The question items included “I would get the vaccine soon”, “If I were to decide whether to get the vaccine today, I would choose to get it”, and “I would get the vaccine in the future”. Answers to the three questions were recorded on a seven-point Likert scale (1 = very little to 7 = very much). The responses were then averaged to create vaccination intention (Mainland: M = 5.27, SD = 1.06, Cronbach’s α = 0.67; TW: M = 4.10, SD = 1.24, Cronbach’s α = 0.84; HK: M = 3.96, SD = 1.44, Cronbach’s α = 0.88).

Demographics, including participants’ age, gender, education, and monthly household income, were included as control variables.

## 3. Results

### Hypotheses Testing

We first conducted correlation analyses of the study variables in three databases (Table 2) with SPSS (IBM, Armonk, NY, USA) and found that most of the study variables, including Media exposure to COVID-19 pandemic, perceived susceptibility and perceived severity of COVID-19, media trust, and vaccination intention, were significantly correlated.

Next, we tested hypothesis 1 with regression in SPSS. The results reveal that media exposure to COVID-19 was significantly associated with individuals’ vaccination intention in all three societies (Mainland: β = 0.32, *p* < 0.001; Taiwan: β = 0.22, *p* < 0.001; HK: β = 0.79, *p* < 0.001). Thus, H1 was supported. The more exposure an individual had to COVID-19 on media platforms, the more likely he or she was to intend to get vaccinated. 

Next, we tested hypothesis 2 with PROCESS macro model 4 and hypothesis 3 with model 14 in SPSS (IBM, Armonk, NY, USA) [100]. PROCESS was more convenient to use and more prevalent in analyzing moderated mediation effects [101].

In response to H2, which concerned the mediating effect of perceived susceptibility and severity of COVID-19 on the relationship between media exposure to COVID-19 and vaccination intention, our results in the Mainland and Taiwan show that both perceived susceptibility and severity of the COVID-19 were significant predictors of vaccination intention (Mainland: susceptibility: β = 0.07, *p* < 0.001; severity: β = 0.13, *p* < 0.001; Taiwan: susceptibility: β = 0.08, *p* < 0.001; severity: β = 0.05, *p* < 0.05). As for Hong Kong, both perceived susceptibility and perceived severity were significant predictors of vaccination intention, but perceived severity was negatively associated with vaccination intention (susceptibility: β = 0.14, *p* < 0.05; severity: β = −0.35, *p* < 0.001). 

The results in Table 3 suggest that the mediating effect of perceived susceptibility and severity was significantly positive in the Mainland and Taiwan (Mainland: susceptibility: *B* = 0.009, SE = 0.003, 95% CI = [0.004, 0.015]; severity: *B* = 0.012, SE = 0.004, 95% CI = [0.006, 0.020]; Taiwan: susceptibility: *B* = 0.011, SE = 0.004, 95% CI = [0.004, 0.021]; severity: *B* = 0.006, SE = 0.003, 95% CI = [0.001, 0.012]). Only perceived severity was a significant mediator of the direct path in Hong Kong (susceptibility: *B* = 0.023, SE = 0.014, 95% CI = [−0.003, 0.053]; severity: *B* = −0.040, SE = 0.010, 95% CI = [−0.062, −0.021]). Thus, H2a and H2b were partly supported. Individuals’ media exposure to COVID-19 would influence their perception of susceptibility and severity, and, in turn, affect their intention to get vaccinated. 

Concerning the moderating effect of media trust (H3), the results in Figure 2 and Figure 3 indicate that media trust moderated the relationship between perceived severity of COVID-19 and vaccination intention in both the Mainland and Taiwan (Mainland: β = −0.06, *p* < 0.001; Taiwan: β = −0.05, *p* < 0.05), but tended to be a non-significant moderator between perceived susceptibility of COVID-19 and vaccination intention (Mainland: β = −0.02, *p* = 0.07; Taiwan: β = 0.03, *p* = 0.21). Conversely, in Hong Kong (Figure 2c), media trust was neither a significant moderator between perceived susceptibility of COVID-19 and vaccination intention (β = 0.12, *p* = 0.06) nor between perceived severity of COVID-19 and vaccination intention (β =−0.03, *p* = 0.66). Thus, H3a was rejected, but H3b was partly supported. When respondents trusted media, their perception of severity would be less likely to influence their intention to get vaccinated.

Moreover, Table 4 suggests a significant moderated mediating effect of severity path in Mainland China and Taiwan (Mainland: *B* = −0.006, SE = 0.002, 95% CI = [−0.011, −0.002]; Taiwan: *B* = −0.006, SE = 0.003, 95% CI = [−0.013, −0.001]). 

Taken together, this model, combining two constructs in the HBM (i.e., perceived susceptibility and perceived severity) and media trust, accounted for 56% of the variance in the intention to receive vaccines among people in Mainland China, 32% among people in Taiwan, and 33% among people in Hong Kong. Thus, this combination based on the HBM largely explained individuals’ vaccination intention in the three societies, especially in Mainland China.

## 4. Discussion

During the global pandemic, media have assumed utmost importance as people have sought to obtain relevant information. The current study revealed that such media exposure would directly and indirectly influence individuals’ health behaviors. With two constructs in the health belief model (i.e., perceived susceptibility and perceived severity) and media trust, this study unfolded its underlying mechanism towards vaccination intention.

By examining the theoretical framework in Figure 1, this study substantiated the direct effect of media exposure on vaccination intention and the mediator role of perceived susceptibility and perceived severity in the direct relationship across all three Chinese societies. Regarding the moderating effects of media trust, there was no influence over the mediating effect of perceived susceptibility. Interestingly, media trust negatively moderated the mediating effect of perceived severity in Mainland China and Taiwan. 

### 4.1. Media Exposure—Golden Ticket to Vaccination Intention 

This study demonstrated a direct relationship between media exposure to COVID-19 and one’s vaccination intention in all three Chinese societies. The more exposure individuals had to media coverage of COVID-19-related information, the more likely they intended to get vaccinated. This finding indicates the potential of media exposure to promote individuals’ preventive behaviors, particularly taking a vaccine in the COVID-19 pandemic. The finding suggests the need for health agencies and governments to provide more COVID-19 information through various media outlets so as to achieve a high vaccination rate that would curb the pandemic.

### 4.2. The Mediating Effects of Perceived Susceptibility and Severity

The current study revealed the utility of perceived susceptibility and severity in the HBM to predict individuals’ vaccination intention (Mainland: 56%; Taiwan: 32%; Hong Kong: 33%). These two constructs were found to be significant mediators of vaccination intention in all three Chinese societies, which is consistent with findings in most of the previous studies (e.g., drug-taking [102]; dental care [103]). 

For Hong Kong, it was interesting to notice that the coefficient of indirect effect through perceived severity was negative. It was partly due to the negative relationship between a high 5.5 index score of perceived severity (M = 5.50) and a low 3.96 index score of vaccination intention (M = 3.96), compared with the other two Chinese societies. Such a negative relationship warrants further investigation with some important societal-level factors (e.g., socio-economic reasons, political reasons, reliability or supply of the vaccine [104]). 

The finding of significant mediating effects with perceived susceptibility and perceived severity expands the development of HBM. Within the context of Chinese societies and disease outbreaks, it refutes the studies by Harrison et al. [28], Carpenter [29], and Janz and Becker [58] that found little predictive power. 

The practical implication is that the government agencies and public health organizations that make efforts to promote vaccinations should appeal to people’s sense of perceived susceptibility and severity. For example, vaccination campaigns can stress the ubiquity of COVID-19 and the potentially disastrous outcomes for individuals who contract it. 

### 4.3. The Surprising Moderating Effect of Media Trust

Although previous literature suggested a positive correlation between media trust and vaccination, its moderating effect was not active with the mediation of perceived susceptibility. It could be considered as a side proof that values such as trust cannot influence people’s evaluation of natural disasters. It would be worthwhile to further examine this supposition with empirical studies.

It came as a surprise that the negative coefficient of media trust over the mediator effect of perceived severity was consistently found in all three Chinese societies, with Mainland China and Taiwan being statistically significant. Such a relationship means that people with a high sense of perceived severity of COVID-19 originally highly intended to get a vaccination. However, the appearance of media trust lessened this relationship. This finding is inconsistent with previous empirical studies examining the effect of media trust on individuals’ outcome behaviors [93].

This outcome may be explained by the framework of agent-structure dichotomy [105]. First, although only media trust was measured in this study, it may implicitly signify people’s general trust in institutions, whether private or public. It is plausible that individuals in these Chinese societies with a relatively collective culture [106,107] were more willing to trust institutions as they might believe that others would take care of them, and it was unnecessary for them to act on anything unless being told to do so [108,109]. This explanation, to a certain degree, was in line with the recent longitudinal study conducted by Wang, et al. [110] in China that found community actions were the key reason for the change of mind on the individual level.

Secondly, another possible interpretation is that the infection rate of the disease (i.e., severity within a nation/community and relative severity compared with other nations/communities) was not severe in the three societies when the surveys were conducted. From August to September 2020, the daily new confirmed cases were all over 0.1 cases per million people in the Mainland, and the highest daily new confirmed cases were 1 case per million people from October 2020 to January 2021 in Taiwan [12]. Since April 2020, the epidemic situation in Mainland China and Taiwan has remained relatively stable, with no large number of newly confirmed cases. As for Hong Kong, the third wave of the pandemic began in early July 2020 and was basically under control by mid-September [111]. In contrast, the global severity of the epidemic was relatively higher. The number of newly confirmed cases worldwide continuously increased from about 30 cases per million in August 2020 to 113.18 cases per million in January 2021. In summary, the severity within three Chinese societies, as well as the relative severity compared with other societies, were generally low during the survey period. Thus, media from these societies might portray a safer environment and control over COVID-19. Under such a circumstance, although individuals understood the pandemic’s negative consequences, compared with those with lower media trust, people with a higher level of media trust will be more likely to develop the idea that it was unnecessary to get vaccinated. 

Therefore, the result of this moderating effect implies that individuals’ level of trust in media might depend on the related contexts. During a pandemic, their trust in media relies on the severity of the pandemic. When situated in a less severe environment, people with higher levels of trust in media will feel it unnecessary to take preventive actions by themselves and be less likely to get vaccinated. 

### 4.4. Limitations and Future Directions 

The study has several limitations. First, the causal relationship cannot be well established with the cross-sectional data at hand. Several waves of the survey should be conducted during the pandemic to enhance the framework’s validity and reliability. Second, although we endeavored to include participants from all ethnic groups in the three societies, the ethnic and linguistic differences may influence participants’ social attitudes. Future studies on societies with various ethnic and cultural groups should take this factor into consideration. Third, this study selected three typical societies to represent Chinese societies as a whole without including some diaspora Chinese living in Southeast Asia. Replication in these societies might be conducted. 

Future studies in these three societies should be cautious about the proportional sample size. Furthermore, when analyzing media effects on individuals’ health decision-making, they should emphasize factors such as government regulation that are intertwined with trust in media, as well as misinformation on media platforms. 

## 5. Conclusions

Overall, the current study adopted the health belief model to explore the relationship between individuals’ media exposure to the COVID-19 pandemic and their intention to receive vaccines. The study derives four major findings. First, it highlighted the significant effect of media on individuals’ vaccine-related decision-making. Second, it justified the utility of the HBM constructs, especially perceived threat (i.e., perceived susceptibility and severity) in predicting individuals’ vaccination intention. Third, by adding media trust as a moderator, this study suggested that the level of trust in media is a contextual factor dependent on the relative severity of the pandemic, i.e., with a relatively lower severity of the pandemic, individuals’ trust in media would impede the influence of perceived severity on their intention to get vaccinated. Fourth, the study examined the direct, indirect, and moderating relationships in the three societies, revealing a general pattern of the relationships in Mainland China and Taiwan, with a slightly different pattern in Hong Kong. 

Additionally, since COVID-19 involved a global pandemic, it is important to overcome the ideological barriers and create effective solutions [112]. Chinese societies, on the whole, may share some of their experience with the rest of the world about COVID-19 vaccination. Specifically, this study found that media were a significant channel for increasing vaccination rates and that individuals’ perceptions of the threat provided a crucial signal of their intent to get vaccinated. Thus, when people perceive a high threat, government promotion of vaccination via media may be more effective. At the same time, purely building collective identities and trust among citizens may be insufficient; context needs to be considered lest efforts to promote vaccination backfire [113]. 

This study represents the first step in evaluating the importance of media across the three Chinese societies. Such analyses can make a tremendous contribution towards scholarship in different fields, including risk management, health communication, and public health. Moreover, the findings of this study provide practical implications for public institutions and health campaign designers. Specifically, governmental agencies and health campaigns may increase the media coverage of vaccine-related information and arouse citizens’ perceptions of COVID-19′s severity, thereby promoting public vaccination intention. 

## Figures and Tables

**Figure 1 ijerph-19-03705-f001:**
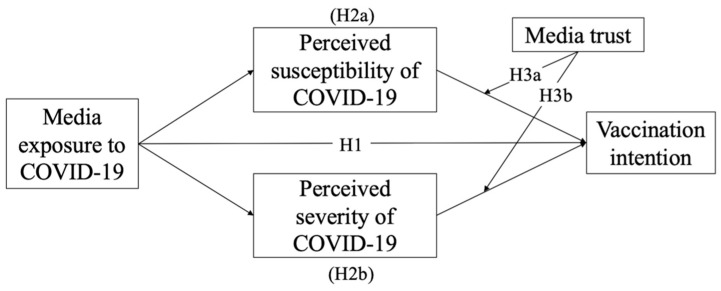
Proposed model of the effect of media exposure on vaccination intention based on the HBM.

**Figure 2 ijerph-19-03705-f002:**
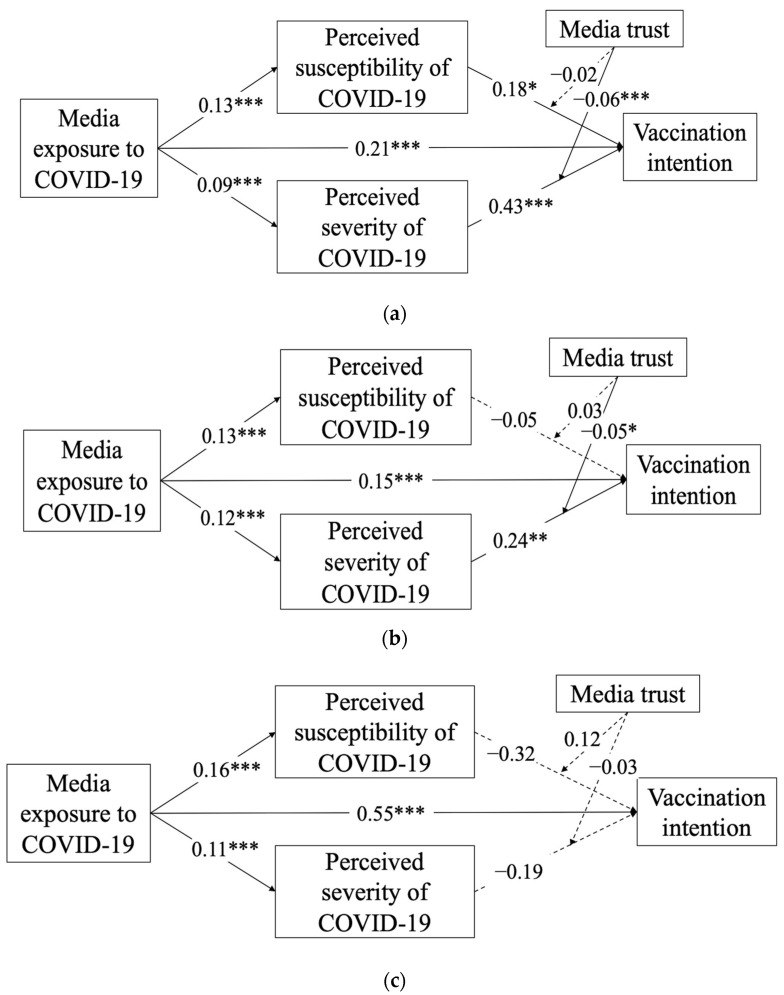
(**a**) Test of the model in Mainland China. (**b**) Test of the model in Taiwan. (**c**) Test of the model in Hong Kong. Note: Solid-line arrows are significant at *p* < 0.05 or higher. Dotted-line arrows are non-significant at *p* < 0.05. * *p* < 0.05; ** *p* < 0.01; *** *p* < 0.001.

**Figure 3 ijerph-19-03705-f003:**
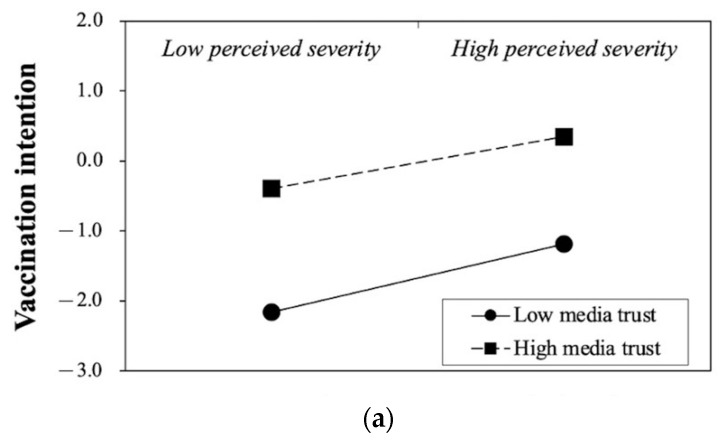
(**a**) Moderating effect of media trust in Mainland China. (**b**) Moderating effect of media trust in Taiwan.

**Table 1 ijerph-19-03705-t001:** Descriptive results in three Chinese societies.

	Mainland	Taiwan	Hong Kong
Gender (male: female)	1:1.004	1:1.278	1:1.128
Age (over 40)	53.3%	29.7%	59%
Monthly household income (Mean)	CNY 15,001–25,000 (USD 2330–3885)	TWD 40,001–50,000 (USD 1424–1780)	HKD 30,001–40,000 (USD 3857–5142)
Education (at least B.A.)	57.5%	71%	35.3%
Top residential areas	Guangdong Province (7%) Anhui Province (6.8%) Jiangsu Province (6.8%)	New Taipei City (20%) Taipei City (14.5%)	New Territories (50%) Kowloon (32.1%) Hong Kong Island (17.5%)

**Table 2 ijerph-19-03705-t002:** Correlation among study variables in three Chinese societies.

	1	2	3	4	5
**Mainland China (*n* = 3389)**					
1. Media exposure to COVID-19 pandemic	-				
2. Perceived susceptibility of COVID-19	0.13 ***	-			
3. Perceived severity of COVID-19	0.10 ***	0.51 ***	-		
4. Media trust	0.39 ***	0.21 ***	0.16 ***	-	
5. Vaccination intention	0.31 ***	0.17 ***	0.25 ***	0.41 ***	-
**Taiwan (*n* = 3000)**					
1. Media exposure to COVID-19 pandemic	-				
2. Perceived susceptibility of COVID-19	0.11 ***	-			
3. Perceived severity of COVID-19	0.07 ***	0.54 ***	-		
4. Media trust	0.29 ***	0.15 ***	0.07 ***	-	
5. Vaccination intention	0.17 ***	0.17 ***	0.13 ***	0.24 ***	-
**Hong Kong (*n* = 3248)**					
1. Media exposure to COVID-19 pandemic	-				
2. Perceived susceptibility of COVID-19	0.14 ***	-			
3. Perceived severity of COVID-19	0.10 ***	0.28 ***	-		
4. Media trust	0.33 ***	0.05 **	−0.03	-	
5. Vaccination intention	0.20 ***	0.03	−0.07 ***	0.25 ***	-

Note: Age, gender, education, and monthly household income were controlled. ** *p* < 0.01; *** *p* < 0.001.

**Table 3 ijerph-19-03705-t003:** Testing the mediating effect of perceived threat on vaccination intention.

Mediators	Perceived Susceptibility				Perceived Severity			
	β	*p*	*B* (Boot SE)	Boot 95% CI	β	*p*	*B* (Boot SE)	Boot 95% CI
Mainland	0.07	< 0.001	0.009 (0.003)	[0.004, 0.015]	0.13	< 0.001	0.012 (0.004)	[0.006, 0.020]
Taiwan	0.08	< 0.001	0.011 (0.004)	[0.004, 0.021]	0.05	< 0.05	0.006 (0.003)	[0.001, 0.012]
HK	0.14	< 0.05	0.023 (0.014)	[−0.003, 0.053]	−0.35	< 0.001	−0.040 (0.010)	[−0.062, −0.021]

Note: Analyses conducted using PROCESS Model 4.

**Table 4 ijerph-19-03705-t004:** Testing the moderated mediating effect of perceived threat on vaccination intention (Media trust as the moderator).

Mediators	Perceived Susceptibility				Perceived Severity			
	β	*p*	*B* (Boot SE)	Boot 95% CI	β	*p*	*B* (Boot SE)	Boot 95% CI
Mainland	−0.02	0.07	−0.003 (0.003)	[−0.009, 0.002]	−0.06	< 0.001	−0.006 (0.002)	[−0.011, −0.002]
Taiwan	0.03	0.21	0.003 (0.004)	[−0.004, 0.012]	−0.05	< 0.05	−0.006 (0.003)	[−0.013, −0.001]
HK	0.12	0.06	0.020 (0.014)	[−0.006, 0.048]	−0.03	0.66	−0.004 (0.010)	[−0.023, 0.015]

Note: Analyses conducted using PROCESS Model 14. The moderating effect refers to perceived susceptibility × media trust and perceived severity × media trust, respectively.

## Data Availability

Due to privacy concerns, we will make the data public two years later.

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
