# Peer review of "A Shot in the Arm for Vaccination Intention: The Media and the Health Belief Model in Three Chinese Societies"

_ijerph, 2022, doi:10.3390/ijerph19063705_

Round 1

Reviewer 2 Report

Interesting study on willingness to take a vaccine in relation to media trust in three different Chinese regions, mainland, Taiwan and Hong-Kong. Media in those three regions is of course different, it is controlled in China, I think controlled in Hong-Kong as well (since the Chinese takeover) and as I understood, in line with political parties in Taiwan. The first thing (as a person from Western Europe) stucks me that those three regions are labelled as regions, whereas you could see at least Taiwan as a separate country.

Is governmental control leading to high scores on media trust? According to the authors it is, whereas I found a comparison where media trust in China is seen as lower than Taiwan (https://ijoc.org/index.php/ijoc/article/view/10141)

Would you as citizen in a controlled governmental surrounding indicate that you don't trust media, or in line with that don't trust the central government? If you risk being arrested and loose your freedom, then I would say that the respondents will always indicate that they trust media. Authors need to take this into account (for instance, where the responses anonymous; if not, then the reliability of the data could be influenced). I do not know how the situation in Hong-Kong is right now, but not mentioning these possible influences is not right I think

Another issue being in the centre of the attention during the pandemic is disinformation regarding vaccines on social media, or even by governmental actors (Trump, Bolsenaro). It is strange to see a paper on health beliefs, media trust related to the covid-19 pandemic, without even the mentioning of disinformation, fake news, vaccine resistance, etcetera. If the authors want to measure and report upon media trust, they should have included information on media in the different regions. Is there a possibility to publish 'anti' information on for instance social media. I guess that is possible in Taiwan, but in the two other countries? Also, are these regions open for non governmental controlled media, like Twitter, Facebook, Instagram etcetera? This might be hard for authors to talk about, but in my view it is essential when media trust is included, or we will get a distorted image. What is media in the three different countries exact? A spokesperson for the government or a free and independent publisher / social media channel, on which you are able to post anything you like, even when it is contrary to what the authorities think or say.

I am very sorry, that I need to reject this paper because of this. I know that this might be hard to be honest about (or maybe you are not allowed to be honest about), but it is in my view essential when we are talking about concepts like media trust.

Some small issues at the end of my review:

  • Table 2: why are you using asterix indicating the significance in mainland China, and give the exact p-value for the other two regions/countries?
  • why don't you present a figure for the moderating effect in Hong-Kong?

Round 2

Reviewer 1 Report

Dear Authors,

I read the revised version of the manuscript with interest. You have made a great job to improve the text. You replied to all my remarks and made even more. The only thing I would like to draw your attention to is the necessary correction of references 31 and 106. They should read:

33. Liu, D.; Komissarov, S.A. Coronavirus as a Political Factor: Western and Chinese Versions of SARS-CoV-2 Origin and the Ways of Its Treatment. The Beacon: Journal for Studying Ideologies and Mental Dimensions. 2021, 4, 020310371. doi: 10.55269/thebeacon.4.020310371

106. Sakwa, R. Pandemic and New Division of the World. The Beacon: Journal for Studying Ideologies and Mental Dimensions. 2021, 4, 010110339. doi: 10.55269/thebeacon.4.010110339

Therefore, I chose "Minor revision" this time. I recommend publishing your paper in IJERPH. And I wish you good luck in your valuable and important scientific work. I am proud of being a reviewer of such interesting work.

Thank you.

Best regards, the Reviewer

Reviewer 2 Report

Thank you for your revisions and your explanations. Very helpful!

I have after rereading your ms a few suggestions. Please consider these as minor suggestions, or advises from my side to improve the paper

  1. I would delete the statistics from the abstract and explain in words the differences
  2. I would add a line on the differences in the time the questionnaires are distributed and the developments of the pandemic. For instance the origin of the Delta variant started in India late 2020, this did not influence your results (because in your regions it was not there) but for the history it might be wise to include this
  3.  
